# Toward Capturing Scientific Evidence in Elderly Care: Efficient Extraction of Changing Facial Feature Points [note 1]

**DOI:** 10.3390/s21206726

**Published:** 2021-10-10

**Authors:** Kosuke Hirayama, Sinan Chen, Sachio Saiki, Masahide Nakamura

**Affiliations:** 1Graduate School of System Informatics, Kobe University, 1-1 Rokkodai-cho, Nada, Kobe 657-8501, Japan; hirayama@ws.cs.kobe-u.ac.jp (K.H.); masa-n@cs.kobe-u.ac.jp (M.N.); 2Department of Data & Innovation, Kochi University of Technology, 185 Miyanigutu, Tosayamada-cho, Kami-shi 782-8502, Japan; saiki.sachio@kochi-tech.ac.jp; 3RIKEN Center for Advanced Intelligence Project, 1-4-1 Nihonbashi, Chuo-ku, Tokyo 103-0027, Japan

**Keywords:** scientific long-term care, facial expression, points of interest, changing point detection

## Abstract

To capture scientific evidence in elderly care, a user-defined facial expression sensing service was proposed in our previous study. Since the time-series data of feature values have been growing at a high rate as the measurement time increases, it may be difficult to find points of interest, especially for detecting changes from the elderly facial expression, such as many elderly people can only be shown in a micro facial expression due to facial wrinkles and aging. The purpose of this paper is to implement a method to efficiently find points of interest (PoI) from the facial feature time-series data of the elderly. In the proposed method, the concept of changing point detection into the analysis of feature values is incorporated by us, to automatically detect big fluctuations or changes in the trend in feature values and detect the moment when the subject’s facial expression changed significantly. Our key idea is to introduce the novel concept of composite feature value to achieve higher accuracy and apply change-point detection to it as well as to single feature values. Furthermore, the PoI finding results from the facial feature time-series data of young volunteers and the elderly are analyzed and evaluated. By the experiments, it is found that the proposed method is able to capture the moment of large facial movements even for people with micro facial expressions and obtain information that can be used as a clue to investigate their response to care.

## 1. Introduction

Japan is facing a super-aged society [1]. The number of elderly people who need care is increasing, which leads to a chronic lack of care resources. Under this circumstance, the Japanese Government has declared the practice of scientific long-term care as a national policy [2,3]. It aims at optimal use of care resources by corroborating the effect of care with scientific evidence. To achieve scientific long-term care, it is essential to evaluate the effect of care objectively and quantitatively. As an objective metric to quantify the care effect, we focus on facial expressions [4,5,6]. The facial expressions give useful hints to interpret the emotions of a person under care. As a practical example, there is a systematic methodology called facial expression analysis [7]. To analyze one’s face and emotions automatically, emerging image processing technologies can be used. Especially, in recent years, owing to powerful machine learning technologies, various companies have provided cognitive application programing interfaces (API) for face [8,9], which automatically recognize and analyze faces within a given picture. These APIs produce values of typical emotions such as happiness and sadness, which would be helpful for the evaluation of the care effect. In actual medical and care scenes, subtle changes of a face that may not be covered by the existing APIs have to be detected. This is because the facial expressions of elderly people under care tend to be weakened due to functional and/or cognitive impairment. Moreover, the emotion analysis conducted by the existing APIs is a black box [10]. Therefore, it is difficult for caregivers to understand why and how the value of the emotion is measured. In addition, analysis of subtle facial changes and explainable emotion analysis become possible, and the user’s expertise in emotion analysis also can be improved.

The main contribution of this paper is to implement a method to efficiently find points of interest from the facial feature time-series data of the elderly. In the existing approach with the facial expression sensing service, since the time-series data of feature values will greatly increase as the measurement time increases, a method to efficiently find points of interest in changes of the facial expressions is desperately needed, especially for evaluating elderly people who can only be shown in a micro facial expression due to facial wrinkles and aging. Analyzing the data obtained, it is possible to analyze them from two perspectives: the macro perspective of how the subject’s emotions changed throughout the care and the micro perspective of when the subject’s emotions changed during care. From the macro perspective, how the care satisfied the subject and how the care suitable for the subject can be considered, and from the micro perspective, it is possible to infer to what kind of conversations or contents the subject responded positively. This kind of analysis is useful for evaluating the effectiveness of care and for providing optimal care according to the preferences of each subject. However, since the facial expression sensing service measures feature values on a second-by-second basis, the time-series data obtained can be enormous, and it may be difficult to find a specific Point of Interest (PoI) [11,12,13], especially when analyzing them from a micro perspective.

The previous version of this paper was published as a conference paper [14]. The most significant change made to this version is the addition of a method for extracting the points of interest from the facial feature time-series data more efficiently. In the proposed method, the moment is regarded as PoIs by us when the subject’s expression changed significantly. To find them, the concept of *Changing Point Detection* is adopted [15,16] and *changing points* automatically detected that indicate the moments at which the feature values fluctuated significantly or the tread of the values changed and find the moments at which the facial expressions of the subject have changed significantly. This makes it possible to easily find PoIs, from the time-series data and provide data that contribute to the analysis from the micro perspective described earlier. Based on the proposed method, test data for evaluation created from the subject’s face and the data measured from the video of subjects receiving care are analyzed using *Change Finder* (i.e., a changing point detection method [17]). Moreover, the effect of the changing point detection method is discussed by comparing it with the actual behavior. In addition, by creating *a composite feature value* [18], which is a combination of more than two feature values instead of single feature values, and applying a changing point detection to it as well, we attempted to obtain information that is closer to fluctuations of actual facial expressions.

The remainder of this paper is organized as follows. Section 2 describes the preliminary and previous study of developing “Face Emotion Tracker (FET)” with a cognitive API. Section 3 introduces the related work of Changing Point Detection and Change Finder from recent years. Section 4 produces a complete description of the proposed method. The preliminary experiment and actual experiment are respectively presented in Section 5 and Section 6, followed by conclusions in Section 7.

## 2. Preliminary and Previous Study

As described in Section 1, to evaluate the effect of care quantitatively and objectively is important for scientific long-term care. However, the evaluation methods of care tend to depend on a subjective scale, such as observations and questionnaires. Therefore, it is difficult to use the obtained data as scientific evidence. Moreover, these assessment methods place a heavy burden on care evaluators, care recipients, or both. In addition, since scientific care is premised on using large-scale data, efficient data collection is required. In the past, to measure care effects objectively, an experiment that measured the changes in the facial expressions of people undergoing care was conducted [19]. In this experiment, facial features such as the height of eyebrows and the opening of eyes were measured from images based on the method called “facial expression analysis” of P. Ekman et al. [7] and so on. However, they measured the features from the recorded video manually. Moreover, in the report, they concluded that “… Currently, we have not generalized this study successfully because of the limited number of cases. … To establish an objective evaluation method, we have to collect and analyze both the data of the objective people and the caregiver more”. From the above, to practice scientific long-term care, it is important to develop a computer-assisted service that enables us to automatically collect data that contribute to the quantitative assessment of care.

With the rapid technological growth of cloud computing, cloud services to process various tasks which were previously executed on edge sides can be used [20]. Under such a kind of background, various services utilizing artificial intelligence, which has become very popular in recent years, are currently available as a cloud service. One of the actual cases of cloud service using artificial intelligence is cognitive services. Cognitive services extract useful information from visual [21], sound [22], linguistic [23], and knowledge data [24] by analyzing them and enable computers to recognize these types of information. Using cognitive services, functions about cognition can be performed which were previously difficult for computers. Many public cloud services provide these kinds of services as “cognitive API” [25]. For example, cognitive APIs for vision can extract various information from an input image [26,27,28]. Cognitive APIs can automatically quantify emotions, which scientific long-term care might utilize.

In the past, our research group developed a real-time emotion analysis service “Face Emotion Tracker (FET)” using a cognitive API and attempted a quantitative evaluation of care [29]. In this previous study, care using a virtual agent [30] was conducted for 5 elderly people living in an elderly facility. During care, FET captures pictures of a subject and sends them to a cognitive API (Microsoft Face API [31]) to analyze the emotion. A part of the result of the FET experiment is shown in Figure 1 as time-series graphs (some outliers has been eliminated). FET outputs 8 kinds of emotions (“anger”, “contempt”, “disgust”, “fear”, “happiness”, “neutral”, “sadness”, “surprise”) from 0 to 1 as probabilities, this time, paying attention to the values of “neutral” and “happiness”. Subject A was relatively expressive, and the values of “neutral” and “happiness” appear alternately during care. Subject B was 99 years old, and the change in facial expression was not so apparent because of the decline of facial muscles. As a result, in contrast to subject A, only neutral was observed at almost all timings. However, the accompanying person who observed the subjects evaluated that subject B felt better and better as the care progressed. Therefore, it can be said that there was a gap between the obtained results and the actual ones. Thus, the reliability of the result of emotion analysis with cognitive API may be degraded depending on the characteristics of the subject.

When observing the facial expressions of care targets, many cases of people with poor facial expressions for their actual emotions will be encountered. The factors involved are aging, dementia [32], the sequela of stroke [33], Amyotrophic Lateral Sclerosis (ALS) [34], Parkinson’s disease [35], and so on. Hence, taking some measures will be needed. In addition, the cognitive API which FET used is built with machine learning technology. Therefore, the process in which a result is outputted from an input cannot be followed. As a result, we may face difficulty when using the obtained data as evidence for the effect of care. To capture scientific evidence in elderly care, facial expressions are regarded as an objective indicator of the effects of care, which led to proposing a user-defined facial expression sensing service in the previous study [14]. The facial expression sensing service extracts the coordinates of characteristic parts of the face (i.e., feature points) such as eyebrows, eyes, and mouth from the face image. Then, the service measures the length connecting two feature points as a *feature value* and records it as time-series data. By tracking the change of these values, the degree of facial expression change from facial movement can be measured. As a result, even when the person shows only a subtle facial expression change, the service will be able to measure and record it as opposed to overlooking it. The aim of this paper is to contribute to explicable emotion analysis and quantitative evaluation of care by examining the degree of facial expression change as clear numerical values and inferring the emotional changes associated with the movement of facial expression.

## 3. Related Work

Since the Facial Expression Sensing Service measures feature values on a second-by-second basis, the time-series data obtained can be enormous, and it may be difficult to find *PoI*, especially when analyzing from a micro perspective. Changing point detection is the technique for detecting changing points in time-series data. This falls into two categories, *offline detection* and *online detection*. Offline detection uses a batch process for data that have been accumulated and find changing points. On the other hand, online detection judges if the data are changing point each time new data are presented. An an example of an offline detection method is a method based on statistical tests by Guralnik and Srivastava [36]. In this method, the point of change in the time-series data is analyzed assuming that all the time-series data to be analyzed are available. In this process, a time-series model has to be applied, such as an autoregressive model or a polynomial regression model, to the previous and next data for all data points, which causes a problem that requires a large amount of computation. In this study, *Change Finder* [17,37,38] (i.e., one of the online change-point detection methods) is adopted. Since the amount of calculation of Change Finder is linear to the number of data to be used, Change Finder can detect changing points quickly [39].

Change Finder is a changing point detection method characterized by an auto regression model (AR model) and two phases of learning using smoothing. Change Finder has a mechanism for detecting changes of the model that the time-series data follow and calculates the degree of change with a changing score. The changing score will be high when the degree of change is high. Moreover, the changing score will be low when the degree of change is low. Change Finder has two AR models. The first model learns original time-series data. The second model learns the degree of changes, which is calculated based on the first model as time-series data. This leads to removing changes raised by small noises. Moreover, using a discounting learning algorithm called Sequentially Discounting AR model learning (SDAR) algorithm to train AR models, Change Finder achieves processing speed as fast as Change Finder can process online. Moreover, whereas AR models require stationarity of data, Change Finder can manipulate non-stationary data by adopting discounting learning. Change Finder processes the one-dimensional time-series data obtained by Facial Expression Sensing Service as follows: **Step 1:** Change Finder trains the first AR model on the input data. In this step, using SDAR algorithm, Change Finder updates the mean, covariance, variance, and autoregression coefficients, which the AR model has. The equations of update are given as follows:(1)$$\mu_{t}\;=\;\left(1-r\right)\mu_{t-1}+rx_{t}$$
(2)$$C_{t,i}\;=\;\left(1-r\right)C_{t-1,i}+r\left(x_{t}-\mu_{t}\right)\left(x_{t-i}-\mu_{t}\right)\;\left(i=0,\ldots ,k\right)$$
(3)$$C_{t,i}\;=\;\sum_{j=1}^{k}a_{t,j}C_{t,i-j}\;\left(i=1,\ldots ,k\right)$$
Here, the order of the AR model is *k*, input data for times t(=1,2,⋯) are $x_{t}$; the average to be calculated this time is $\mu_{t}$; the auto regression coefficient is $a_{t,i}$; the covariance is $C_{t,i}$; the variance is $\sigma_{t}$, and the discounting rate is *r*(0<r<1).

Equation (Equation 3) is called the Yule–Walker equation. In this regard, let $C_{t,-i}\;=\;C_{t,i}$. By solving this equation, $a_{t,j}$ is obtained, and then, the following is calculated:(4)$${\hat{x}}_{t}\;=\;\sum_{i=1}^{k}a_{t,i}\left(x_{t-i}-\mu_{t}\right)+\mu_{t}$$
(5)$$\sigma_{t}\;=\;\left(1-r\right)\sigma_{t-1}+r\left(x_{t}-{\hat{x}}_{t}\right)\left(x_{t}-{\hat{x}}_{t}\right)$$
 **Step 2:** In this step, Change Finder calculates the score using normal probability density distribution based on average and variance that results from Step 1. Let the probability density be $p_{t}\left(x_{t}\right)$, then $p_{t}\left(x_{t}\right)$ and the score $y_{t}$ are calculated as follows.
(6)$$p_{t}\left(x_{t}\right)\;=\;\frac{1}{\sqrt{2\pi \sigma_{t}}}\exp\left(-\frac{1}{2\sigma_{t}}{\left(x_{t}-{\hat{x}}_{t}\right)}^{2}\right)$$
(7)$$y_{t}=-\log p_{t-1}\left(x_{t}\right)$$
 **Step 3:** In this step, Change Finder smoothes scores that result from Step 2 to remove noises.
(8)$$Score_{t}\;=\;\frac{\sum_{i=0}^{w-1}y_{t-i}}{w}$$
Here, the width of smoothing is *w*, and the result of smoothing is $Score_{t}$.

 **Step 4:** Using $Score_{t}$ that results from Step 3 as new time-series data, Change Finder performs the second round of learning, calculating the score and smoothing just as in Step 1–Step 3. Then, scores that result from the second round of Step 3 as the changing score are obtained.

## 4. Methodology

### 4.1. Purpose and Approach

In this study, the analysis focuses on the micro-perspective described in Section 1 and aims to efficiently investigate when the subject reacted during care. For this purpose, we consider extracting *PoI*, as the moments when large fluctuations are recognized in the time-series data of feature values obtained by Facial Expression Sensing Service.

A simple way of extracting PoIs is to look for points whose feature values exceed a certain threshold. However, this method considers the absolute values of the feature values, even if a subject’s expression has changed significantly, for example, when a neutral face becomes a smiley face, those facial changes may be discarded if the change in feature values is small. Moreover, since feature values are based on facial movements, the values are always fluctuating little by little. Therefore, depending on the threshold setting, a large number of points that exceed the threshold may be extracted, and thus the efficiency of analysis will not be improved. Considering these factors, it cannot be said that this method can capture facial expression changes based on reality, and extracted points will not be so valuable as PoIs. Therefore, in this study, the change detection algorithm called Change Finder described in Section 3 and attempts to extract PoIs based on accurately capturing the change in the feature values are adopted.

The key idea of this study is to introduce the concept of *composite feature value* and apply change-point detection to it as well as to single feature values. Composite feature value is a scale created from multiple feature values according to a certain purpose of the analysis. In this way, analysis can be performed more efficiently than considering the changes in each feature value. Furthermore, since multiple facial parts move when facial expression changes, the composite feature value that combines multiple feature values will enable the extraction of PoIs with high validity that reflect the actual facial movements. The specific method is as follows.

 **Step 1:** Create a composite feature value $X\left(t\right)\;=\;f(x_{1}\left(t\right),x_{2}\left(t\right),\cdots )$ by combining multiple features $x_{1}\left(t\right),x_{2}\left(t\right),\cdots$ **Step 2:** Input X(t) into Change Finder and obtain the change score S(t). **Step 3:** Set a threshold γ, and regard a point $t^{\prime }$ with $\gamma <S(t^{\prime })$ as a candidate of PoI.

### 4.2. Composite Feature Value

In this study, from the perspective of measuring the effectiveness of care, we consider defining “*Smile scale*” as a composite feature that indicates how much the subject enjoys the care. In creating Smile scale, facial movements suggesting feelings of enjoyment and happiness are considered. Based on Ekman et al.’s book on facial expression analysis [7] and the attempt to observe the features of faces showing various emotions taken with the cooperation of volunteers [40], the following features of facial expressions showing happiness are summarized. From this description, this study focuses on 4 of the 6 feature values used in the experiments of preliminary study (see Figure 2) as the smile scale: “Mouth width” $h_{9}$, “Eyebrow height” $a_{1}$, “Eyes opening” $a_{3}$, and “Mouth corner lowering” $a_{4}$. Out of the four features that we focus on, when compared with the features of facial expressions described earlier, the three features except for the “Mouth width” $h_{9}$ show more happiness when the value is *smaller*. Moreover, the absolute amount of variation differs for each feature value. For example, the movement around the mouth should be larger than the movement of eyes opening and closing. Therefore, it is necessary to assign certain weights to each feature.

Eyebrows will be lowered in whole;Eyes narrowed;Mouth corner will be pulled back and raised;Cheeks will be raised.

To determine the weights, we first measured the feature values of neutral and smiling faces, using the same face images of this thesis author as in the preliminary experiment. Next, we randomly selected 10 data in each state and calculated the average of the feature values and differences between the two states. The samples of the face images used in the calculations are shown in Figure 3 for the neutral state and Figure 4 for the smiling state. The results are shown in Table 1. The “Ratio of difference” shows the ratio of the “Difference” of each feature value compared to the “mouth width” $h_{9}$. The results show that the absolute amount of change in “Eyebrow height” $a_{1}$ and “Eyes opening” $a_{3}$ is small, about half of that of “Mouth width” $h_{9}$ and “Mouth corner lowering”. $a_{4}$. Therefore, Smile scale *X* is determined as follows.
(9)$$X\;=\;2h_{9}-a_{1}-a_{3}-2a_{4}$$

Considering that the degree of facial movement varies among individuals, using Ratio of change in Table 1 directly as a coefficient was avoided. Note that the standardized single feature values to calculate *X* was used. Specifically, for the original feature value x(t), which has the mean $\mu_{x}$ and standard deviation $\sigma_{x}$, z(t) is calculated by the following formula and is used as the feature value:(10)$$z\left(t\right)\;=\;\frac{x\left(t\right)-\mu_{x}}{\sigma_{x}}$$

### 4.3. Experiment Detail

We attempted to extract PoIs for single feature values and composite features value Smile scale, using Change Finder on the test data created from the face of this thesis’ author and data obtained from facial images of elderly people receiving care. In this experiment, we use the Change Finder library for Python [41]. The library takes the following parameters: order of the AR model order, discounting rate *r*, and width of smoothing size *w*. These parameters are used directly for the first AR model training. In the second AR model training, the width of the smoothing size will be round(*w*/2.0). In each experiment, the parameters are set to order=1, r=0.025, and w=5. Among the change scores S(t) calculated by the Change Finder, as shown in Figure 5, we define the point *t* with the highest score as the time between the score exceeding a certain threshold and it falling below the threshold as the *PoI candidate*. By examining the appearance of the face around PoI candidates, we can confirm whether an obtained PoI candidate is a valid as PoI. In this series of experiments, we determine the threshold γ by $\gamma \;=\;\mu_{S}+\sigma_{S}$ using the mean $\mu_{S}$ and standard deviation $\sigma_{S}$ of the change score *S*.

## 5. Preliminary Experiment

### 5.1. Outline

As a preliminary experiment, we conducted a PoI extraction experiment using the feature value data obtained from the face image of the author of this paper was conducted. The test data based on a video in which the author repeated a neutral and a smile expression several times was created. The video was taken by using the built-in camera of a laptop computer while the author was looking at the screen. We measured the feature values at intervals of 1 s and removed obvious outliers. Missing data were filled with previous data. Then Change Finder was applied to the single features and Smile scale of this data to calculate change score and extract PoI candidates.

In this experiment, the ratios $n_{p}/n$ and $n_{o}/n$ were first calculated to estimate how much more efficient the facial expression analysis becomes. Here, *n* is the number of whole data; $n_{p}$ is the number of PoIs, and $n_{o}$ is the number of data exceeding the threshold γ. The smaller these values are, the fewer the number of data to be checked, and the more efficient the analysis will be. Next, to confirm the validity of the obtained PoI candidates, the timing of facial expression changes recorded by eye observation was compared with them. First, we looked for the timing, as shown in Figure 6, when the face (a) becomes happy looking and (b) changes to not happy looking by visual inspection, and recorded them as “*event point*”. In this experiment, event points were recorded only when obvious facial movement was seen. For example, gradually returning from a smiling face to a neutral face over a long period, about 30 s to 1 min was not recorded because it was difficult to determine where to place an event point. Next, PoI candidates were picked whose timing matched an event point, and we defined it as a *correct PoI* and defined its number as a *number of correct*. In this experiment, if a PoI candidate was within 10 s before or after an event point, it was considered that they are matched. Note that a number of correct will be counted only once for each event point. Then, for the number of PoIs $n_{p}$, the number of correct $n_{c}$, and the number of event points $n_{e}$, the following ratios were checked.

**Recall**: Ratio of event points that could be captured $n_{c}/n_{e}$;**Precision**: Ratio of valid PoI candidates $n_{c}/n_{p}$.

The higher the Recall, the less the change in facial expression is missed. Moreover, the higher the Precision, the more accurate the extraction of PoI is with less noise. The data at the beginning of the change score are not useful because the training of the SDAR algorithm has just started. Therefore, the data in the first 30 s from the calculation of the threshold γ and each ratio were excluded.

### 5.2. Results

Table 2 shows the calculation results of each ratio. Figure 7 shows the change score for Smile scale. The red line is the change score, the light blue line is the raw value of each feature value, and the black dashed line is the threshold γ. Red dots indicate PoI candidates. Among them, dots with double circles indicate correct PoIs. The colored vertical lines indicate the event points, where yellow lines indicate when the subject’s face became happy looking and light blue lines dashed line indicates when the subject’s face became not happy looking. The left axis indicates the value of the change score, the right axis indicates the feature value, and the horizontal axis shows the elapsed time. $n_{p}/n$ was less than 0.02 in all cases, and $n_{o}/n$ was about 0.1 at the highest. In other words, our method was able to narrow down the data to be searched to less than one-tenth. When Change Finder was applied to the smile scale, the number of correct was the highest and Recall was also the highest. Looking at the graphs, PoI candidates near the event points at around 2 min, 3 min, and 15 min and 30 s were obtained only from the smile scale, and these were not obtained for any single feature value. In addition, for the case of Smile scale, the number of PoI candidates not around event points was the lowest, so the Precision was the highest. From these results, it can be said that PoI candidates of composite feature value have a higher probability to catch important moments than PoI candidates of single feature values.

In the graphs of each feature value, there are PoI candidates in the 6 to 9 min where there are no event points. Although at these points the author does not show smiles, subtle facial movements were observed such as widening eyes and holding the mouth. Change Finder seemed to react to these facial movements. As long as the data are based on detailed facial movement tracking, this kind of noise will occur more or less. However, when using Smile scale, the subtle variation in each feature value is suppressed a little. Therefore, it is considered that Change Finder extracts few unimportant points compared to the case using single feature values, and as a result, the Precision is increased.

As a comparison with the method of this experiment, the PoI candidates were extracted and each scale calculated using the raw values of the Smile scale. As in the case in which the change score was used, the timing with the highest value exceeding a certain threshold was considered as a PoI candidate. We set the threshold value to μ+nσ using the mean μ and standard deviation σ of all Smile scale’s raw values. The results are shown in Table 3. When we set the low threshold (n=0, n=0.5), the values of $n_{p}/n$ and $n_{o}/n$ were much higher than those using the change score. Moreover, many PoI candidates were detected, and although Recall was high, Precision was low. When we set the high threshold (n=2), the number of correct decreased and the precision was low because small changes were ignored. In the case of n=1, although Recall was relatively high, many PoI candidates were detected in the areas where Smile scale value wiggled, and the precision was lower than that using the change score. From this, it can be said that using the change score is more efficient and accurate than analyzing the raw feature data while changing the threshold value.

From this result, it was confirmed that applying Change Finder to feature value data and using composite features are effective for the test data. In the next section, we will conduct the same analysis using actual images of elderly people during care and verify whether the analysis method of this preliminary experiment can be applied to general data.

## 6. Actual Experiment

### 6.1. Overview

In this experiment, we measure a feature value using images of five subjects during care obtained in the experiment of FET [29] and attempt to extract PoIs on them using Change Finder as in the preliminary experiment. Then, we examine whether the analysis can be more efficient and whether the obtained PoI candidates are valid. We divided expressive subjects (subjects A, C, and D) and subjects with poor facial expression (subjects B and E) into subject group 1 and subject group 2, respectively, and discussed each group. Since the face images of each subject were taken at intervals of about 3 s, the obtained feature value had similar intervals. To adjust the experimental conditions with the preliminary experiment, in advance, we filled the feature value data with their previous data, to place the data at intervals of 1 s. Moreover, we removed obvious outliers. As in the preliminary experiment, we excluded the data in the first 30 s from the calculation of the threshold γ and from each ratio.

### 6.2. Results

Here, we explain the results of the analysis for subject group 1. In addition to calculating the change score of Change Finder, for Subject Group 1, since we could examine facial expression changes visually, as in the preliminary experiment, we recorded event points and calculated Recall and Precision. Table 4, Table 5 and Table 6 show the calculation results of each ratio for each subject. Figure 8, Figure 9 and Figure 10 are the graphs showing the change scores of the Smile scale. The legends of each graph are the same as that of the preliminary experiment. For each subject, $n_{o}/n$ was about 0.08 at the highest for any of the feature values. Considering that the data to be investigated can be reduced to less than 8% of the total data, it can be said that the analysis was efficient. In this experiment, the original data were heavily padded, so $n_{p}/n$ is not very helpful, but the values were all less than 0.02.

For subjects C and D, the highest values of Recall and Precision were obtained when using the Smile scale. Moreover, for both subjects, there were event points that could be detected only when using the Smile scale. From these results, it can be said that using the composite feature value was useful for subjects C and D, as the preliminary experiment. In both subjects, Recall was 0.5, which means that half of the event points were detected as PoI candidates. The graph of the Smile scale shows that although the value rises and falls around the event points, not all of them were extracted as PoI candidates. In other words, it can be considered that Change Finder does not simply react to all local changes of feature value. However, considering that the data to be investigated was greatly narrowed down and that the extracted PoI candidates corresponded to the event points to some extent, it can be said that although there were some missing event points, Change Finder was able to efficiently scoop up the appropriate timing as PoI from the constantly changing feature data. As for the subject A, compared to Smile scale, “Mouth width” has a higher value in both Recall and Precision, and “Mouth corner lowering” has a higher value in Recall. Looking at the graph of Smile scale, compared to the large changes in feature values around 4 min, 10 min 30 s, and 14 min, the change score did not react to the small value changes between 6 min 30 s and 7 min, and corresponding event points were overlooked. On the other hand, they were detected when using “mouth width”. Although the results of single feature values are more likely to catch unimportant points as PoI candidates than composite feature values, they can be considered to be effective in complementing the results of composite feature value depending on the characteristics of the input data.

We describe the results of the analysis for subject group 2. For subject group 2, it was difficult to find event points visually, so we only calculated the change score of Change Finder and discussed it, confirming the facial expression around detected PoI candidates The calculation results of $n_{p}/n$ and $n_{o}/n$ are shown in Table 7 and Table 8 for each subject. Figure 11 and Figure 12 is the graphs showing the change scores of Smile scale. The legends of the graphs are the same as those of the preliminary experiment and the experiment of subject group 1. For each subject, $n_{o}/n$ was always less than 0.1. As in the case of subject group 1, it can be said that this method was able to narrow down the data significantly. Subject B had hardened facial muscles and no obvious change in facial expression compared to the other subjects. Although each feature value continued to fluctuate, it was not possible to identify the timing when the values rose or fell clearly or when the trend of the time-series changed. Around the PoI candidate using single feature values, it was possible to confirm the facial movements corresponding to each feature value, but there were no clear signs that emotional changes can be inferred. Examining the faces near the PoI candidates using the smile scale, we mainly observed mouth movements such as holding the mouth and opening the mouth. While there was no particular tendency of variation in every single feature, the Smile scale might be strongly reflected by “Mouth width” and “Mouth corner lowering”, which have a high coefficient in the Smile scale, and this made Change Finder sensitive to these fluctuations. Subject E showed symptoms of euphoria and generally appeared to be happy from the beginning to the end of the care. When we examined the faces near the PoI candidate extracted from the smile scale, we were able to capture the moments when the expressions moved significantly. In the first half to the middle part of the care (before 14 min), large smile scenes could be seen. In the latter part of the care, there was a candidate point of interest at 18 min and 30 s, and from this time, facial movements including smiling were not seen very much until about 22 min and 30 s. According to the care record, the subject was listening to her favorite music for about 4 min from 18 min and 20 s, and it can be assumed that her facial expression temporarily faded as a result of listening to the music intently.

### 6.3. Discussion

For subject group 1, we confirmed that the extracted PoI candidates matched the visually confirmed event points, (i.e., the change in timing of facial expression, to some extent). Although there were gaps in accuracy depending on the data used and some overlooked event points, this method seems to be useful in that it can narrow down a huge amount of time-series data and provide clues to efficiently find PoIs. In common with the three subjects of subject group 1, closely examining the graph of the change score, we found some small increase in the score around the event point, although it does not exceed the threshold γ. If the threshold is lowered, such points will be detected, but the number of PoI candidates will also increase, and the cost to examine appropriate PoI will increase. Therefore, it will be necessary for users to determine the threshold value by considering their own needs, the time and cost they can take for analysis, and so on. Regarding the composite feature value, looking at the results for the Smile scale of subject group 1, the failure to detect event points when the face changes to not happy looking is more noticeable than when the face becomes happy looking. For example, in the case of Subject D’s Smile scale, the former was detected in 5 out of 7 event points (There is one event point of happy looking on the right edge of the graph), while the latter was detected only in 2 out of 6 event points. The Smile scale was created to estimate how happy subjects feel. However, since the facial movements corresponding to emotions such as happiness, disgust, and surprise are unique, it would be necessary to create a composite feature value for each emotion, especially if we want to capture the “unpleasant look”. For subject group 2, we could not successfully extract PoI candidates that showed a large change in facial expression from those whose facial movements were small. However, we could see some facial movements like mouth movements around the PoI candidates, so even though it is difficult to estimate emotions, it is possible to estimate whether the interaction was successful without losing the subject’s interest by investigating the “magnitude of response to external stimulus”. On the other hand, for the subject who seemed to be euphoric, the moments when she showed a big smile or became serious were successfully captured as PoI candidates.

We have conducted experiments based on the images of the subject during care obtained in the FET experiment. In this section, we attempt to extract the PoI candidates using the original measurement results of the FET experiment and compare the results with those obtained by using the feature values obtained by the Facial Expression Sensing Service. FET can quantify various kinds of emotions (see Section 2). In this experiment, we focus on “happiness” from the viewpoint of estimating the effect of care. For the time-series data of the “happiness” value, as in the previous experiments, the time-series data at 1-s intervals and the extracted PoI candidate were interpolated using Change Finder with the settings described in Section 6.1. We calculated $n_{p}/n$ and $n_{o}/n$ for each subject. The results are shown in Table 9. For subject group 1, we calculated Recall and Precision. The results are shown in Table 10. Moreover, the graphs of results of subjects in each subject group are also shown in Figure 13 and Figure 14. The right vertical axis of each graph shows the value of “happiness”. Other than that, legends are the same as in the previous experiments.

In this experiment, $n_{o}/n$ was lower than 0.1 for all subjects, which means that we were able to focus on the important data, significantly, the FET data as well as the data from the Facial Expression Sensing Service. In subject group 1, for subjects A and D, Recall and Precision were higher than those with feature values. For subject C, the result was not good. If the data tend to keep high or low values, as in the case of subject C, the subtle changes will be lost and PoIs will not be detected well. For subject group 2, the change score run along with the change in the value of “happiness”. The value of “happiness” is almost always around 0 or 1, and it tends to change sharply at some moment. Therefore, such instantaneous value is mainly detected as a PoI candidate, and the detailed changes in facial expressions at other moments are lost. Moreover, we compared the change scores and PoI candidates for each subject in subject group 2 with those using the Smile scale. The result are shown in Figure 15. From these graphs, it can be seen that there are some gaps in the timing of the PoI candidates between the two cases. The gaps are especially seen in subject B. This suggests that by using the data from the Facial Expression Sensing Service, it is possible to perform different analyses of the case using FET data. From these results, we conclude that the analysis based on the method of this experiment can be used for FET data for expressive people. For the people with poor facial expression, we will be able to conduct care analysis based on subtle facial changes that cannot be captured by FET by using the data of the Facial Expression Sensing Service.

## 7. Conclusions

In order to capture scientific evidence in elderly care, this paper specifically focused on the changes in the facial expressions of the elderly. The main outcomes include the following: (1) Proposing to apply the Changing point detection method using Change Finder with the feature value data obtained from Facial Expression Sensing Service to efficiently find PoIs. (2) Conducting a complete experiment to analyze and evaluate the performance of the proposed method whether on the facial expressions of young men or elderly people. Through the experiments, it was found that the proposed method cannot only efficiently find appropriate PoIs from a huge amount of time series data but also extract the moments when the face moved remarkably. Meanwhile, for processing the special case with incomplete raw data to the maximum, improvement and verification of the proposed method are required. As future work, more facial data from elderly people will be analyzed and evaluated integrating the proposed method and care nursing knowledge, in order to realize explainable scientific care for elderly people.

## Figures and Tables

**Figure 1 sensors-21-06726-f001:**
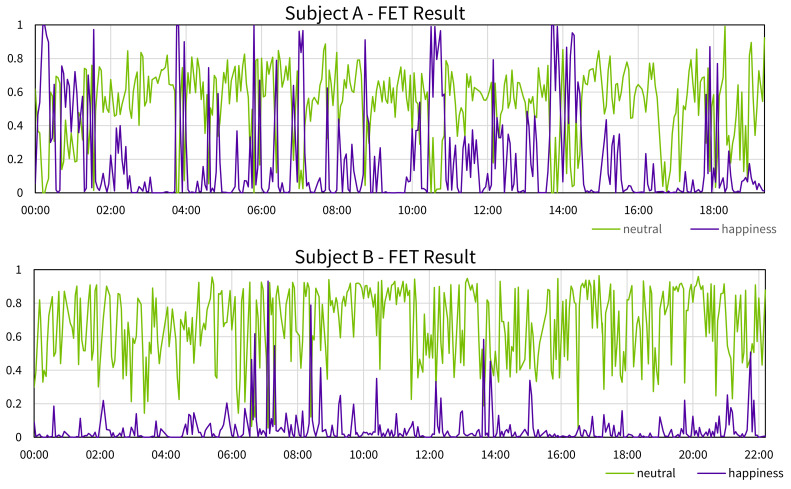
A part of the result of the FET experiment.

**Figure 2 sensors-21-06726-f002:**
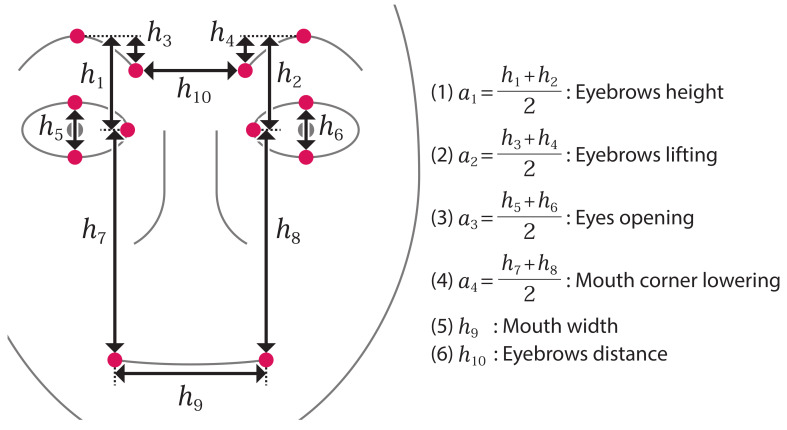
Feature values for the experiment.

**Figure 3 sensors-21-06726-f003:**
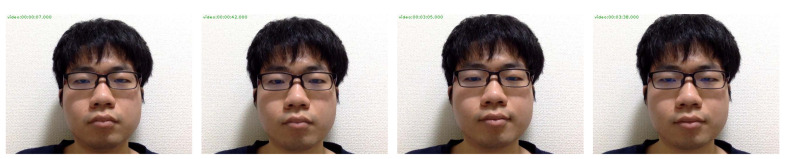
Samples of neutral face images to consider smile scale.

**Figure 4 sensors-21-06726-f004:**
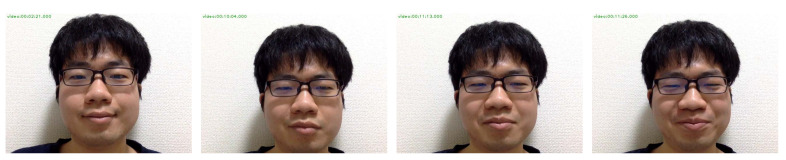
Samples of smiling face images to consider smile scale.

**Figure 5 sensors-21-06726-f005:**
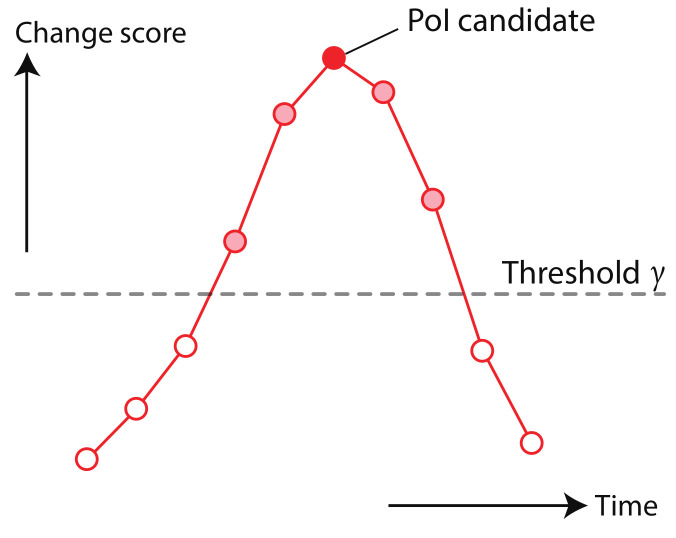
Definition of PoI candidate.

**Figure 6 sensors-21-06726-f006:**
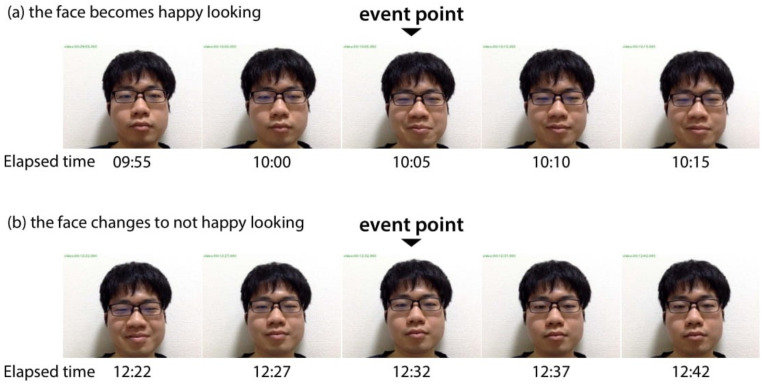
Example of event points (**a**) the face becomes happy looking, (**b**) the faces changes to not happy looking. (“Elapsed time” corresponds to change score graphs in Figure 7).

**Figure 7 sensors-21-06726-f007:**
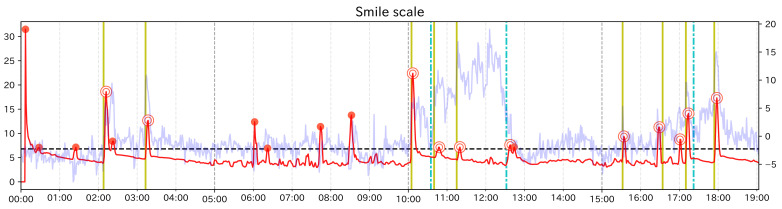
The graph showing the change score for Smile scale (test data).

**Figure 8 sensors-21-06726-f008:**
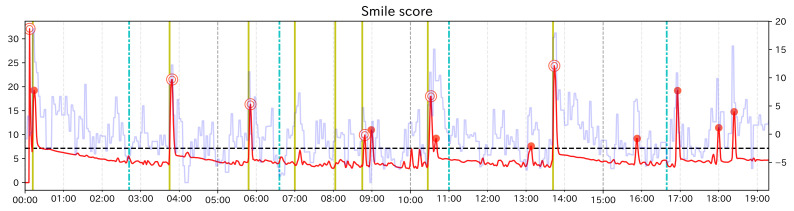
Graph of the change score for Smile scale (Subject A).

**Figure 9 sensors-21-06726-f009:**
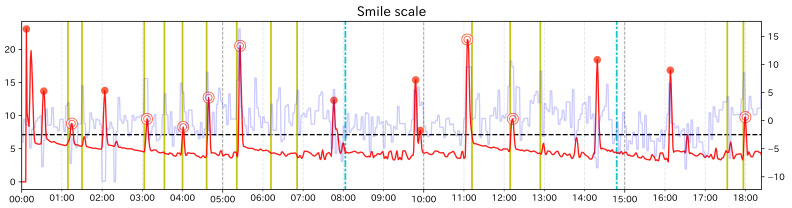
Graph of the change score for Smile scale (Subject C).

**Figure 10 sensors-21-06726-f010:**
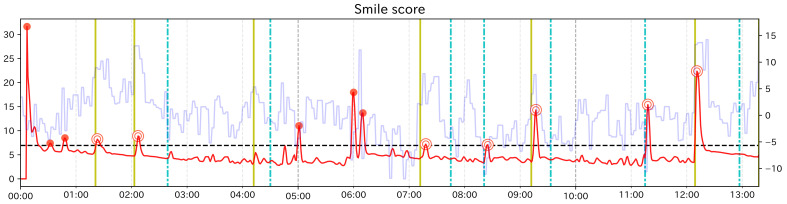
Graph of the change score for Smile scale (Subject D).

**Figure 11 sensors-21-06726-f011:**
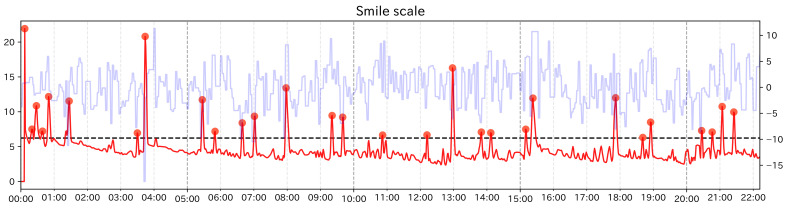
Graph of the change score for the Smile scale (Subject B).

**Figure 12 sensors-21-06726-f012:**
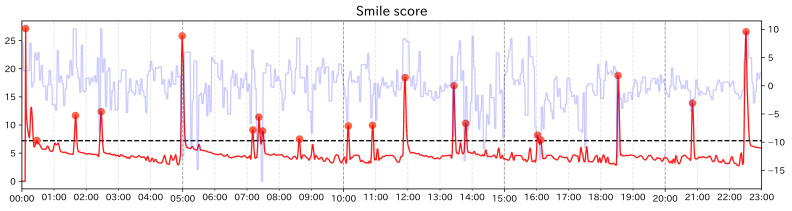
Graph of the change score for the Smile scale (Subject E).

**Figure 13 sensors-21-06726-f013:**
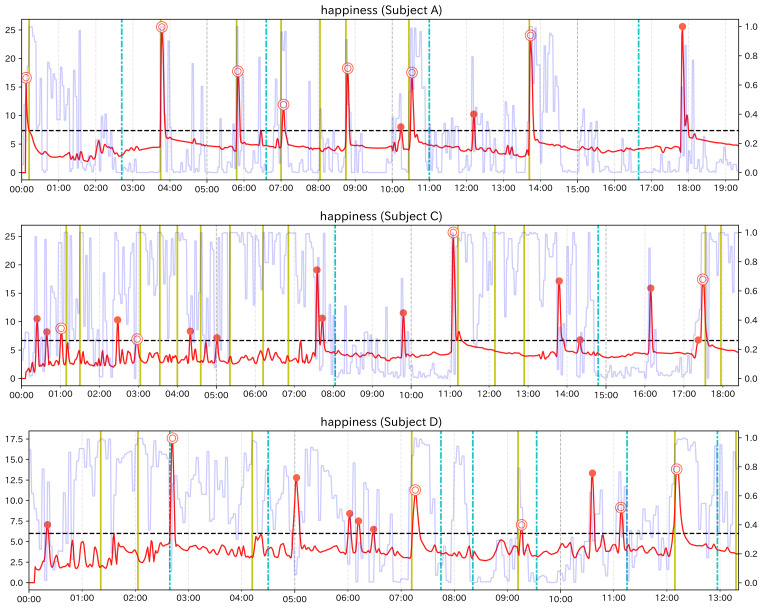
Graphs of the change score for FET’s “happiness” value (Subject group 1).

**Figure 14 sensors-21-06726-f014:**
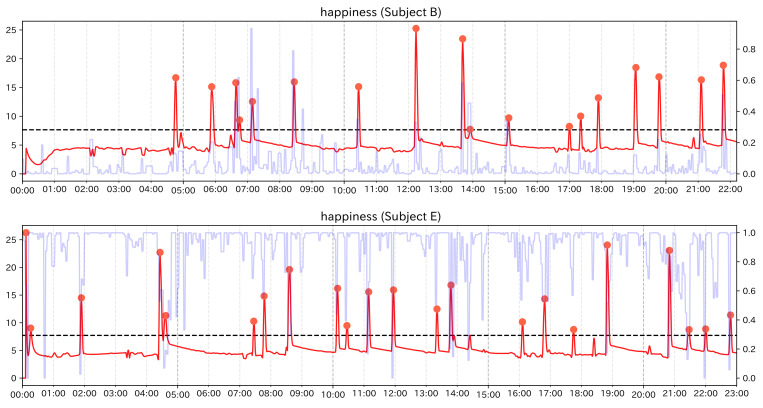
Graphs of the change score for FET’s “happiness” value (Subject group 2).

**Figure 15 sensors-21-06726-f015:**
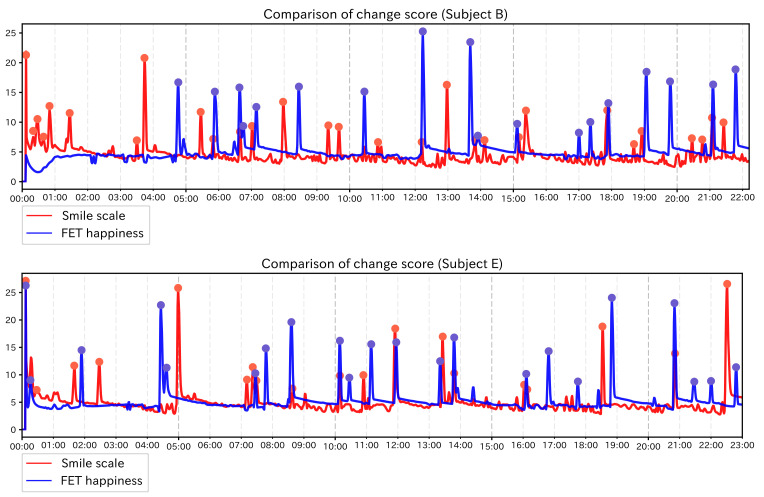
Comparison of change score between Smile scale and FET “happiness” value (Subject group 2).

**Table 1 sensors-21-06726-t001:** Average of 10 feature values for the neutral and the smiling faces of the author.

	Mouth Width $h_{9}$	Eyebrows Height $a_{1}$	Eyes Opening $a_{3}$	Mouth Corner Lowering $a_{4}$
Average of neutral	0.8199	0.4270	0.0901	1.0431
Average of smiling	0.8914	0.3981	0.0503	0.9817
Difference	0.0715	−0.0289	−0.0398	−0.0614
**Ratio of difference**	**1.0000**	**0.4035**	**0.5561**	**0.8578**

**Table 2 sensors-21-06726-t002:** Various parameters and ratios measured (test data: number of whole data *n* = 1114, number of event points $n_{e}$ = 12).

Feature Type	Single				Composite
Feature Name	Mouth Width $h_{9}$	Eyebrows Height $a_{1}$	Eyes Opening $a_{3}$	Mouth Corner Lowering $a_{4}$	Smile Scale
$n_{p}$	16	12	16	13	18
$n_{o}$	55	80	75	113	78
$n_{p}$/*n*	0.0144	0.0108	0.0144	0.0117	0.0162
$n_{o}$/*n*	0.0494	0.0718	0.0673	0.1014	0.0700
$n_{c}$	4	3	4	6	11
Recall	0.3333	0.2500	0.3333	0.5000	0.9167
Precision	0.2500	0.2500	0.2500	0.4615	0.6111

**Table 3 sensors-21-06726-t003:** Various parameters and ratios of the experiment measured using the raw Smile scale value (test data: number of whole data *n* = 1144, number of event points $n_{e}$ = 12).

Threshold	μ	μ + 0.5σ	μ + σ	μ + 2σ
$n_{p}$	69	24	22	11
$n_{o}$	355	227	173	77
$n_{p}$/*n*	0.0603	0.0210	0.0192	0.0096
$n_{o}$/*n*	0.3103	0.1984	0.1512	0.0673
$n_{c}$	8	8	9	5
Recall	0.6667	0.6667	0.7500	0.4167
Precision	0.1159	0.3333	0.4091	0.4545

**Table 4 sensors-21-06726-t004:** Various parameters and ratios measured (Subject A: number of whole data *n* = 1129, number of event points $n_{e}$ = 11).

Feature Type	Single				Composite
Feature Name	Mouth Width $h_{9}$	Eyebrows Height $a_{1}$	Eyes Opening $a_{3}$	Mouth Corner Lowering $a_{4}$	Smile Scale
$n_{p}$	13	17	19	20	12
$n_{o}$	61	76	86	93	62
$n_{p}$/*n*	0.0115	0.0151	0.0168	0.0177	0.0106
$n_{o}$/*n*	0.0540	0.0673	0.0762	0.0824	0.0549
$n_{c}$	6	3	3	7	5
Recall	0.5455	0.2727	0.2727	0.6364	0.4545
Precision	0.4615	0.1765	0.1579	0.3500	0.4167

**Table 5 sensors-21-06726-t005:** Various parameters and ratios measured (Subject C: number of whole data *n* = 1075, number of event points $n_{e}$ = 16).

Feature Type	Single				Composite
Feature Name	Mouth Width $h_{9}$	Eyebrows Height $a_{1}$	Eyes Opening $a_{3}$	Mouth Corner Lowering $a_{4}$	Smile Scale
$n_{p}$	12	12	15	10	15
$n_{o}$	67	45	77	61	76
$n_{p}$/*n*	0.0112	0.0112	0.0140	0.0093	0.0140
$n_{o}$/*n*	0.0623	0.0419	0.0716	0.0567	0.0707
$n_{c}$	6	4	6	4	8
Recall	0.3750	0.2500	0.3750	0.2500	0.5000
Precision	0.5000	0.3333	0.4000	0.4000	0.5333

**Table 6 sensors-21-06726-t006:** Various parameters and ratios measured (Subject D: number of whole data *n* = 769, number of event points $n_{e}$ = 14).

Feature Type	Single				Composite
Feature Name	Mouth Width $h_{9}$	Eyebrows Height $a_{1}$	Eyes Opening $a_{3}$	Mouth Corner Lowering $a_{4}$	Smile Scale
$n_{p}$	10	8	7	6	12
$n_{o}$	42	38	36	61	54
$n_{p}$/*n*	0.0130	0.0104	0.0091	0.0078	0.0156
$n_{o}$/*n*	0.0546	0.0494	0.0468	0.0793	0.0702
$n_{c}$	5	2	2	3	7
Recall	0.3571	0.1429	0.1429	0.2143	0.5000
Precision	0.5000	0.2500	0.2857	0.5000	0.5833

**Table 7 sensors-21-06726-t007:** Various parameters and ratios measured (Subject B: number of whole data *n* = 1303).

Feature Type	Single				Composite
Feature Name	Mouth Width $h_{9}$	Eyebrows Height $a_{1}$	Eyes Opening $a_{3}$	Mouth Corner Lowering $a_{4}$	Smile Scale
$n_{p}$	23	19	19	21	26
$n_{o}$	95	86	88	129	115
$n_{p}$/*n*	0.0177	0.0146	0.0146	0.0161	0.0200
$n_{o}$/*n*	0.0729	0.0660	0.0675	0.0990	0.0883

**Table 8 sensors-21-06726-t008:** Various parameters and ratios measured (Subject E: number of whole data *n* = 1351).

Feature Type	Single				Composite
Feature Name	Mouth Width $h_{9}$	Eyebrows Height $a_{1}$	Eyes Opening $a_{3}$	Mouth Corner Lowering $a_{4}$	Smile Scale
$n_{p}$	14	16	20	14	17
$n_{o}$	72	70	100	71	82
$n_{p}$/*n*	0.0104	0.0118	0.0148	0.0104	0.0126
$n_{o}$/*n*	0.0533	0.0518	0.0740	0.0526	0.0607

**Table 9 sensors-21-06726-t009:** Values of $n_{p}/n$ and $n_{o}/n$ from change score using FET “happiness” data.

	*n*	$n_{p}$	$n_{o}$	$n_{p}$/*n*	$n_{o}$/*n*
Subject A	1132	9	54	0.0080	0.0477
Subject B	1303	18	84	0.0138	0.0645
Subject C	1075	15	70	0.0140	0.0651
Subject D	772	10	53	0.0130	0.0687
Subject E	1351	20	90	0.0148	0.0725

**Table 10 sensors-21-06726-t010:** Recall and precision from change score using FET “happiness” data of Subject Group 1.

	$n_{e}$	$n_{c}$	$n_{p}$	Recall	Precision
Subject A	11	6	9	0.5455	0.6667
Subject C	16	4	15	0.2500	0.2667
Subject D	14	5	10	0.3571	0.5000

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
