# Peer review of "Toward Capturing Scientific Evidence in Elderly Care: Efficient Extraction of Changing Facial Feature Points†"

_sensors, 2021, doi:10.3390/s21206726_

Round 1

Reviewer 1 Report

While the current manuscript is well presented in terms of structure and figures and the proposed application is a field in which research is important to help with the elderly care not only in Japan but in may aging nations around the world, there are a few significant issues regarding the significance of the contribution proposed in this paper.

Firstly in terms of the need of this method I am not convinced it is necessary, if part of the key idea is to find the time when the emotion changed, could this simply not be achieved by a simple timestamp on an frame when the emotion detection system, Most video based emotion detection would normally process only a small number of frame for real-time performance and can detect when emotional change takes place as the predicted emotion of the sub-set of frames would change, I am not clear from the research why the method proposed is significant or performs something that has not already been achieved previously in the many are of Facial Emotion Recognition (FER) fields?

Furthermore, the results present do not show a high level of accuracy, this is especially true for the second group with diminished level of facial motion which is the key demographic being target in the introduction of the research, where only good levels of detection happen with large facial deformation (euphoria), but not smaller motions.

Either further results benchmarked against some form other FER methodology and/or clear arguments to show why this is significant and needed would much improve this paper.

There are also changes that could be made to improve the reading this includes but is not limited to:

Line 10 – “By the experiments, it is clear that the proposed method can be performed more efficiently than 11 considering the changes in each feature value.” This is ambiguous  

Line 45 - Consider changing “to find point of interest (PoI)” to “to find a specific Point of Interest (PoI)”

Line 50 – Need rewording as not clear English:  It has a huge potential in finding the points of interest  timely care and reduce a heavy burden on data analysis by the manual

Line 57 – What is ‘my face’, do you mean one of the authors or is this a capture tool?

Line 75 Change “expressions of object people” to “expressions of people”

Line 97: Remove the =

Line 108: PoI has already been defined earlier in the manuscript

Line 119: “of online” to “of the online”

Line 120: remove “,” and replace with “and”, also no space between last work and the reference [28].

Line 144: change “reacted in the care” to “reacted during care” and again PoI has already been defined no need to do this multiple times.

Line 143: Has a link to 1. Should this say section 1 as this is where it links to.

Line 187: “the face images of this thesis author used in the after-mentioned preliminary experiment” change to something like “the face images of the author previously used in the preliminary experiment”

Line 194: Try to make the equation follows this text instead of the table if possible.

Line 282: change “likewise” to “like”

Reviewer 2 Report

Dear Authors

The manuscript is somehow interesting, the language is not OK, the paper must be reviewed and the quality should be enhanced. The Reviewer is really concerned about the novelty of the work. It is quite vague!

The paper has been structured well. However, there are too many subsections correspond to each main section. It is recommended to merge the subsections and simplify them.

However, in order to enhance the manuscript quality and to the best attention, please address the following comments in the revised version.

  1. What is the main novelty of the work? Although it was stated in the abstract, it seems vague to the readers. Please clearly describe the main contribution of the work to the state of the art.
  2. From technical point of view, it is suggested to use the passive tense instead of active tense. Therefore, please avoid using the sentences beginning with “we”. In 111 places, sentences were found. Please correct it
  3. How did you develop the mathematics? Are they taken from the literature? Please cite a reference for them.
  4. What is your intention to present section 2 as “ 2. Related Work”? Is it a part of literature review? If so, you must rearrange the introduction section and the literature review section.
  5. There are too many figures! I do recommend the authors to reduce the number of figures by merging them together and simplify as much as possible. Figures were presented in good quality.
  6. Conclusions are not supportive! It must be rewritten in a way that the main outcome of this work presented as bullets.

Very Best

The Reviewer

Round 2

Reviewer 1 Report

The revised manuscript is much improved and the initial comments have been answered, great job.  There are still a few typos that need to be corrected in this version these include:

Please check the document there are many lines including 356, 358, 375, 376 where there looks to be a space between the word and the punctuation that needs to be removed. There also seems to be some double spaces but that could just be the way the document is displayed.

Line 447: change capital letter in eXplsinable.

Reviewer 2 Report

Dear Authors

Good improvement!

Accept!